# Awakening City: Traces of the Circadian Rhythm within the Mobile Phone Network Data

Gergő Pintér * and Imre Felde 

John von Neumann Faculty of Informatics, Óbuda University, Bécsi út 96/B, 1034 Budapest, Hungary; felde.imre@uni-obuda.hu
* Correspondence: pinter.gergo@nik.uni-obuda.hu

**Abstract:** In this study, call detail records (CDR), covering Budapest, Hungary, are processed to analyze the circadian rhythm of the subscribers. An indicator, called wake-up time, is introduced to describe the behavior of a group of subscribers. It is defined as the time when the mobile phone activity of a group rises in the morning. Its counterpart is the time when the activity falls in the evening. Inhabitant and area-based aggregation are also presented. The former is to consider the people who live in an area, while the latter uses the transit activity in an area to describe the behavior of a part of the city. The opening hours of the malls and the nightlife of the party district are used to demonstrate this application as real-life examples. The proposed approach is also used to estimate the working hours of the workplaces. The findings are in a good agreement with the practice in Hungary, and also support the workplace detection method. A negative correlation is found between the wake-up time and mobility indicators (entropy, radius of gyration): on workdays, people wake up earlier and travel more, while on holidays, it is quite the contrary. The wake-up time is evaluated in different socioeconomic classes, using housing prices and mobile phones prices, as well. It is found that lower socioeconomic groups tend to wake up earlier.

**Keywords:** mobile phone data; call detail records; type allocation code; data analysis; human mobility; urban mobility; social sensing; socioeconomic status; circadian rhythm; sleep–wake cycle



## 1. Introduction

The mobile phone network, during its operation, constantly communicates with cell phones. This communication can be divided into two categories: (i) the passive, cell-switching communication that keeps the cell phones ready to use the mobile phone network at any time, and (ii) the active, billed usage of the mobile phone network, including phone calls, text messages or mobile internet usage. The call detail records (CDR) collect the latter, containing information about the subscriber, the time of the activity and the place (via the cell), where the activity occurs.

In the last few decades, anonymized CDR have become a standard information source for analyzing the characteristics of human mobility. The billed activities of the subscribers are recorded, providing information about the whereabouts of the population. The human mobility analysis, based on this massive information source, is utilized in such fields—among others—as social sensing, epidemiology, transportation engineering, urban planning or sociology. Furthermore, the human sleep–wake cycle (SWC) is also studied by analyzing mobile phone network data.

CDR processing is often applied for large social event detection, such as football matches [1–4], concerts [5], sociopolitical events [6,7] or mass protests [8,9]. Epidemiology is mentioned as a potential application of human mobility studies, but the COVID-19 pandemic prioritized its applications in digital epidemiology, as mobile phone network data can reflect the mobility changes caused by the imposed restrictions. Willberg et al. found a considerable decrease in the population presence in the largest cities of Finland, after the lockdown [10].

Romanillos et al. reported similar results from the Madrid metropolitan area [11]. Lee et al. examined the mobility changes during the lockdown in England, and found that the mobility of the wealthier subscribers decreased more significantly [12]. Khataee et al. compared the effect of the social distancing in several countries, using mobility data from Apple iPhones [13]. Bushman et al. [14], Gao et al. [15], Hu et al. [16] and Tokey [17] also analyzed effects of the stay-at-home distancing on the COVID-19 increase rate in the U.S. Lucchini et al. studied the mobility changes during the pandemic in four U.S. states [18].

Identifying the home and work locations of a subscriber is a common [19–27] and crucial part of the CDR processing, as these locations fundamentally determine the people's mobility customs. Furthermore, a good portion of the people live their lives in an area that is determined by only their home and workplace [19,21] or their communities [28]. Using the home and work locations, the commuting trends can be analyzed [24,25,27,29,30].

Analyzing city structure led to the analyses of the socioeconomic structure of the population, as different social classes live in different parts of a city, but CDRs also have been used to analyze gender and minority segregation [31]. Xu et al. [23] used six mobility indicators, housing prices and per capita income in Singapore and Boston to analyze the socioeconomic classes. It was found that the wealthier subscribers tend to travel shorter distances in Singapore, but longer ones in Boston. Barbosa et al. also found significant differences in the average travel distance between the low and high income groups in Brazil [32]. In a previous work, we also demonstrated differences in mobility customs between socioeconomic classes [27] in the case of Budapest. Ucar et al. revealed the socioeconomic gap by mobile service consumption [33]. Vilella et al. found that education and age play a role in news media consumption patterns in Chile, using a dataset that provides information about the visited websites [34].

The studies, cited before, mainly focus on the spatial distance between the home and work locations, as it is hard to estimate the travel time using sporadic CDR data, though it has a seasonal nature due to the human biorhythm. Moreover, human mobility is highly regular [35,36], and the individual activity has a bursty characteristic [37]. Jo et al. found that by removing the circadian and weekly seasonality, the bursty nature of the human activity remains [38].

In the digital era, the human sleep–wake cycle (SWC) is also studied using info-communication systems, such as smartphones [39,40], websites [34,41,42], social media [43] and call detail records (CDR) [38,44–49]. Cuttone et al. used screen-on events of the smartphones to study the daily sleep periods [39], and Aledavood et al. examined the social network of different chronotypes, using the same data set [40]. Monsivais et al. identified yearly and seasonal patterns in calling activity and resting periods [47,48]. Lotero et al. found a connection between temporal patterns and the socioeconomic status of the subscribers, namely that the wealthier wakes up later [46]. Diao et al. found a difference in the daily activity between different districts of Boston [50].

Economic models distinguish city parts, such as residential areas, industrial areas, business districts and so on, but that is a rather static, slowly evolving layer of a city. Mobile phone network data have a potential to describe the city structure via the inhabitants' mobility patterns. This study focuses on the effect of the SWC to the city structure. In this regard, it is a continuation of our previous work [27], but this time, the city structure is analyzed by the circadian rhythm of the people who live and work in a given area of Budapest.

Is it possible to cluster the areas of a city by time, when the activity of the inhabitants, the workers, or the passers-by start their activity in the morning or halt in the evening? Do city parts have "chronotypes"? Can neighborhoods or districts be described by the terms "morningness" or "eveningness"? Is there a structural or socioeconomic connection between the areas with the same "chronotype"?

The goal of this study is to answer these questions, and the contributions of this paper are briefly summarized as follows:

1. Introducing wake-up time as an indicator to describe the behavior of a group of subscribers.
2. Using this indicator to distinguish the areas of Budapest.
3. Estimating the day length and the working hour length, using the same method.
4. Demonstrating connection between the wake-up time and the mobility customs.
5. Identifying correlation between the wake-up time and the socioeconomic status.

The rest of this paper is organized as follows. The utilized data are described in Section 2. Then, in Section 3, the applied methodology is introduced, and in Section 4 the results of this study are presented and discussed. Finally, in Section 5, the findings of the paper are summarized, and a conclusion is drawn.

## 2. Materials

Vodafone Hungary, one of the three mobile phone operators providing services in Hungary, provided two anonymized CDR data sets for this study. The observation area was Budapest, the capital of Hungary and its agglomeration. Both observation periods are one month long: the first is June 2016, the second is April 2017. The nationwide market share of Vodafone Hungary was 25.3% in 2016 Q2, and 25.5% in 2017 Q2 [51].

The communication between a cellular device and the mobile phone network can be divided into two categories: (i) an administrative communication maintaining the connection with the service, for example, registration of the cell switching that can be called passive communication; and (ii) when the device actively uses the network for voice calls, message or data transfer, that can be called active communication. The available data contain only the active communication, which is sparser, so they cannot be used to track continuous movements.

The obtained data contain a hashed value to identify the SIM (subscriber identity module), a timestamp that is truncated to 10 s, and an ID of the cell. Thus, a subscriber can be mapped to a geographic location in a given time. These are extended with the type of the customer (business or consumer), the type of the subscription (prepaid or postpaid), the age and gender of the subscriber and the type allocation code (TAC) of the device. The TAC is the first eight digits of the IMEI (international mobile equipment identity) number, and allocated by the GSM Association and uniquely identifies the mobile phone model. These values are also present in every record, so, for example, the device changes can be tracked as well.

Both the type of the activity (voice call, message, and data transfer) and the direction (incoming and outgoing) were omitted from the data, and there were no data provided by the operator to resolve the TACs to manufacturer and model.

The data were processed using the same framework as in our previous works [4,27]. During the data cleaning process, the wide format was normalized. The CDR table contained only the SIM ID, the timestamp, and the cell ID. For the subscriber and subscription-related information, a separate table was formed, and another table was created to track the device changes of the subscriber.

Another table is created for the cell information, including a cell ID, the geographic location of the cell centroid and base station. The cell centroid is an estimation based on a momentary state, especially with the UMTS cells that apply a load-balancing mechanism, called "cell breathing", which can change the geographic size of the serving area. The heavily loaded cells shrink, and the neighboring ones grow to compensate [52].

### 2.1. June 2016

This data set includes 2,291,246,932 records from 2,063,005 unique SIM cards, during the observation period of June 2016. Figure 1 shows the activity distribution between the activity categories of the SIM cards. The dominance of the last category, SIM cards with more than 1000 activity records, is even more significant. It can be seen that almost 27% of the SIM cards produce more than 91% of the total activity.

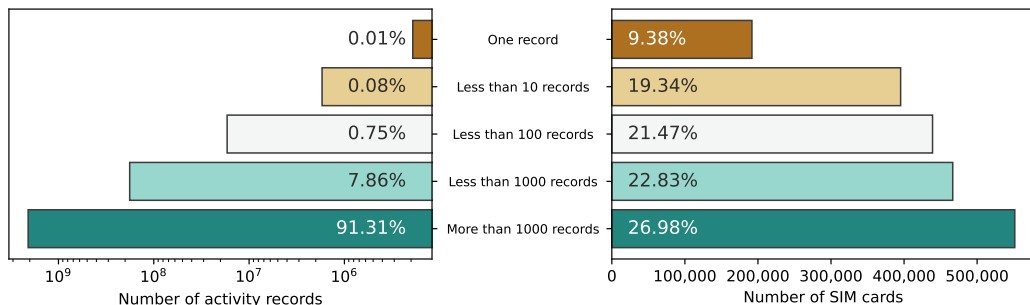

**Figure 1.** The SIM cards, in the "June 2016" data set, categorized by the number of activity records. The SIM cards with more than 1000 activity records (26.98% of the SIM cards) provide the majority (91.31%) of the activity.

Figure 2 shows the SIM card distribution according to the number of active days. Only 34.59% of the SIM cards had activity on at least 21 different days. In total, 241,824 SIM cards (11.72%) appeared on at least two days, but the difference between the first and the last activity was not more than seven days. This may indicate the presence of tourists. High levels of tourism are usual during this part of the year.

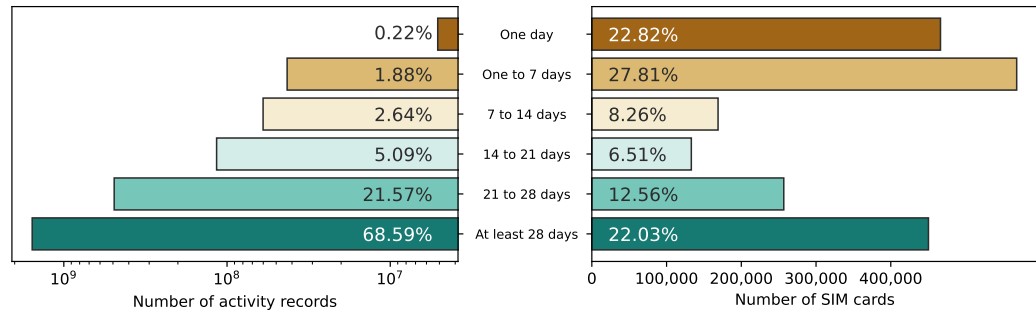

**Figure 2.** SIM card distribution of the "June 2016" data set, by the number of active days.

Type allocation codes are provided for every record because a subscriber can change their device at any time. Naturally, most of the subscribers (95.71%) used only one device during the whole observation period.

While the subscription details were available for every SIM card, the subscriber information was missing in slightly more than 40% of the cases, presumably because of the subscribers' preferences regarding the use of their personal data.

Although the data contained cell IDs, only the base station locations, where the cell antennas were located, were known. As a base station usually serves multiple cells, these cells were merged by the serving base stations. After the merge, 665 locations (sites) remained with known geographic locations. To estimate the area covered by these sites, Voronoi Tessellation was performed on the locations. This is a common practice [24,29,53–55] in CDR processing.

### 2.2. April 2017

The second CDR data set contains 955,035,169 activity records, from 1,629,275 SIM cards, and the observation period is April 2017. Figure 3 shows the activity distribution between the activity categories of the SIM cards. Only 17.67% of all the SIM cards, that have more than 1000 activity records, provide the majority (75.48%) of the mobile phone activity during the observation period. Figure 4 shows the distribution of the SIM cards by the number of active days. Only about one-third (33.23%) of the SIM cards have activity on at least 21 different days. In spite of the relatively large number of SIM cards present in the data, most of them are not active enough to provide enough information about their mobility habits. Figure 5a shows the mobile phone network activity distribution during the observation period, and Figure 5b its Fourier decomposition to highlight the seasonality of

the data. As expected, this 30-day dataset has a 24 h periodicity. In this study, mainly this data set is used, and the "June 2016" is only used as a reference for some analyses (e.g., in Section 4.2).

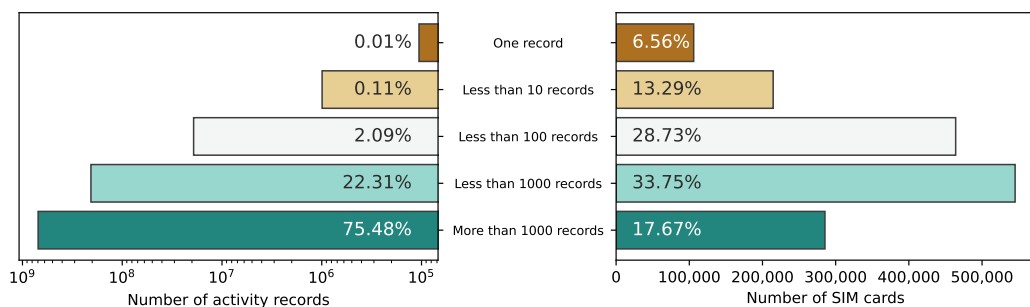

**Figure 3.** The SIM cards in the "April 2017" data set categorized by the number of activity records. The SIM cards over a thousand records (17.7%) provide the majority (75.48%) of the activity.

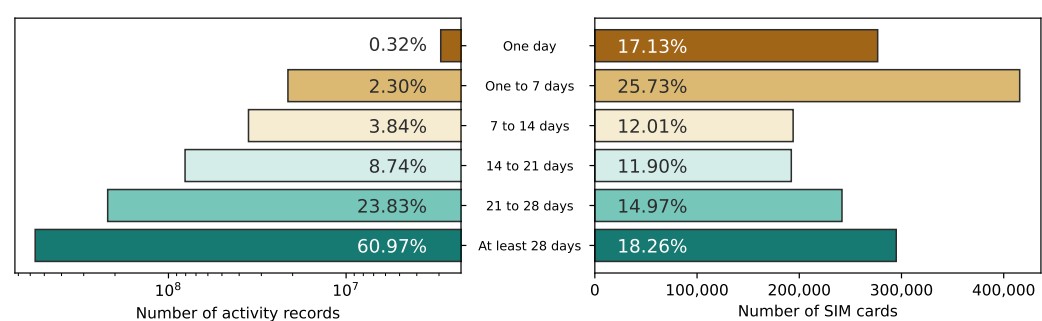

**Figure 4.** SIM card distribution of the "April 2017" data set, by the number of active days.

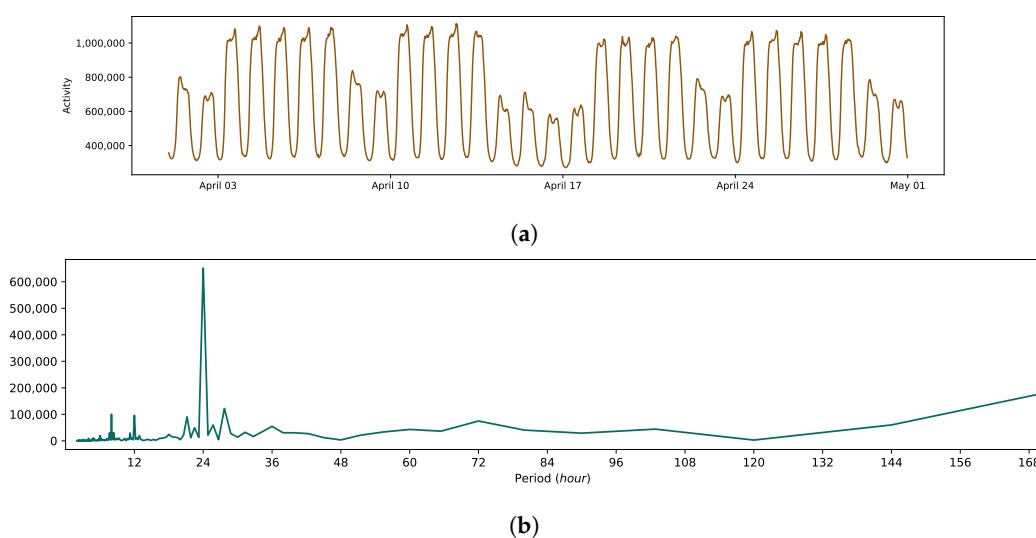

(**a**)

(**b**)

**Figure 5.** The mobile phone network activity (**a**) during the observation period of April 2017, and its Fourier decomposition (**b**).

### 2.3. Data for the Socioeconomic Indicators

In a previous work [27], real estate prices were used to characterize the socioeconomic status. The ingatlan.com estate selling website provided more than 60 thousand estate locations, as a database snapshot from August 2018, with floor spaces and selling prices from Budapest and Pest county. These prices were used to describe the property prices at the subscribers' home locations. Figure 6a shows the distribution of the normalized housing prices from the advertisements.

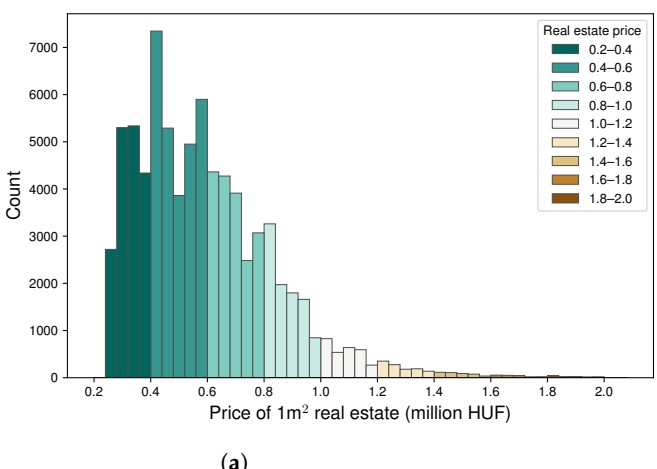

(**a**)

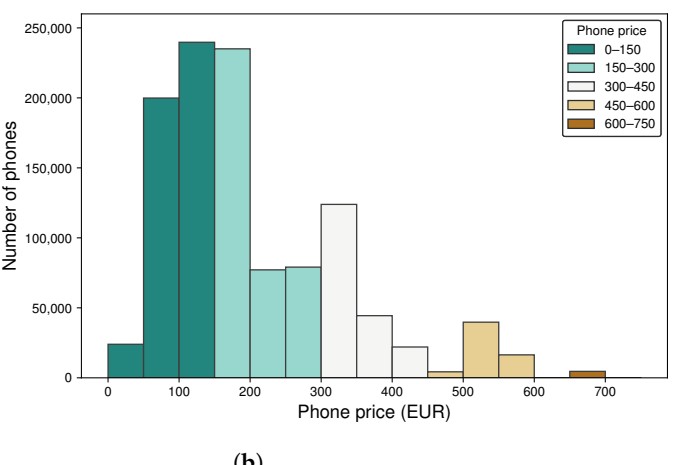

(**b**)

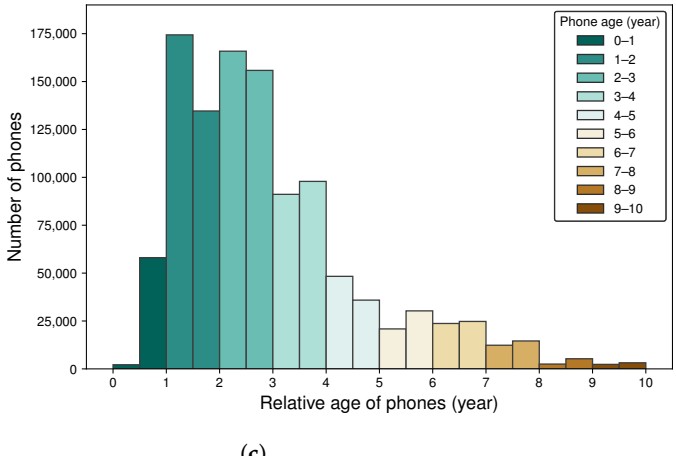

(**c**)

**Figure 6.** Distribution of the real property prices (**a**), the mobile phone prices (**b**), and the mobile phone relative ages (**c**).

Our former paper [4] focused on the price, and the relative age of the subscribers' mobile phone were utilized as a socioeconomic indicator. The data, used for resolving the TACs to the hardware vendor and model, were provided by 51Degrees. With the model and type of the device in which the SIM cards operate, they can also be used to remove those

SIM cards that are not used in mobile phones, and thus, do not represent a human. The cell phone prices and the release dates were obtained from GSMArena [56]. The relative age of a phone is determined as the difference of the date of the CDR data set (e.g., April 2017) and the release date of the phone. Figure 6c shows that most of the devices are one to three years old, but there are some very outdated (over eight years) phones in use. Figure 6b displays the distribution of the cell phone prices. Although, the data source is supposed to contain launch prices, the validation analysis (Appendix B) implies that the phone prices are depreciated.

## 3. Methodology

In a previous work [27], a framework was introduced to process the mobile phone network data, described in Section 2. The CDRs were normalized, cleaned and the mobility metrics (Section 3.2) were determined for every subscriber.

### 3.1. Home and Work Locations

Most of the inhabitants in cities spend significant time of a day at two locations: their homes and work places. In order to find the relationship between these most important locations, first, their positions of these locations have to be determined. There are a few approaches used to find home locations via mobile phone data analysis [20,22,25,26,57].

Our solution is similar to the most common approach. The most frequent cell where a SIM card is present during working hours is considered the work location, on workdays between 09:00 and 16:00. The home location is calculated as the most frequent cell where a SIM card is present during the evening and night on workdays (from 22:00 to 06:00) and all day on holidays. Although people do not always stay at home on the weekends, it is assumed that a significant amount of activity is generated from their home locations. In [27], we used census data to indirectly, via commuting trends, validate the estimated home and work locations.

### 3.2. Mobility Metrics

Along with the home and work locations, the radius of gyration and the entropy are commonly used [18,19,23,25,27,53,58,59] indicators of human mobility, and were also determined for every subscriber.

The radius of gyration [35] is the radius of a circle in which an individual (represented by a SIM card) can usually be found. It was originally defined in Equation (1), where $L$ is the set of locations visited by the individual, $r_{cm}$ is the center of mass of these locations, and $n_i$ is the number of visits to the $i$-th location.

$$r_g = \sqrt{\frac{1}{N} \sum_{i \in L} n_i (r_i - r_{cm})^2} \tag{1}$$

The entropy characterizes the diversity of locations visited using an individual's movements, defined as Equation (2), where $L$ is the set of locations visited by an individual, $l$ represents a single location, $p(l)$ is the probability of an individual being active at a location $l$, and $N$ is the total number of activities taken part in by an individual [53,58].

$$e = -\frac{\sum_{l \in L} p(l) \log p}{\log N} \tag{2}$$

### 3.3. Wake-Up Time

The call detail records are aggregated for every cell and 10 min time intervals. Then, the moving average is applied with the window of 12. The minimum of the aggregated records is usually in the middle of the night, and the maximum is in the afternoon. To determine the wake-up time, the positive edge of the curve needs to be detected, when the number of the mobile phone activity increases drastically in a short period of time. The time that is considered the wake-up time is simply where the activity value reaches the

mean of the minimum and the maximum of the curve. Figure 7 illustrates the concept. Equation (3) defines the mean of the activity edge, where $a_{min}$ and $a_{max}$ are the minimum and the maximum of the (daily) activity level, respectively.

$$m = (a_{max} - a_{min}) * 0.5 + a_{min} \tag{3}$$

This process is repeated for every day, then the median of the daily wake-up times are determined. Analogously to the wake-up time, the bedtime can be calculated to describe when the mobile phone activity decreases significantly, as selecting the mean value on the negative edge. It has to be noted that these values are naturally not the actual times when people wake up or fall asleep. Those moments cannot be determined using simply the mobile phone network. Using the screen-on events of the phone [39] can be much closer to the actual values, especially in the case of the wake-up time, if the phone is used as an alarm clock. In spite of this, it is supposed that this approximating method can reveal the rough tendencies of the daily routine. Still, the terms "wake-up time" and "bedtime" are used to refer to the times of the positive/rising and negative/falling edges of the daily activity curve, respectively.

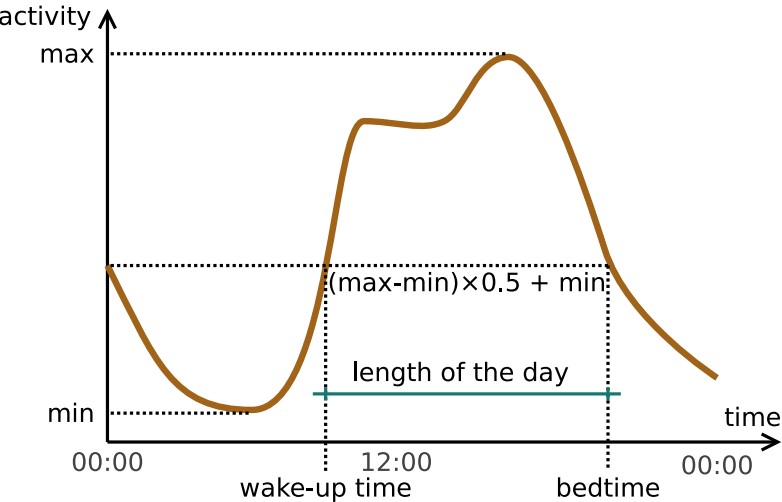

**Figure 7.** Calculation of the wake-up time and the bedtime.

### 3.4. Aggregation of the Subscribers

As the type of the phone activity is unknown, the wake-up time of the individual is hard to estimate. Since making a call requires being awake (whereas a message can be received and data can be transferred autonomously), knowing which activities represent phone calls would provide more accurate information about when people are certainly awake. Instead of this, the significant activity increase is to be used to identify the wake-up time.

Furthermore, the activity of a single device is also sporadic and does not provide enough data to identify an activity increase for every day. To mitigate the downside of the sporadic nature of the data, the devices are grouped, which can be performed in two ways: (i) calculate the wake-up time of an area (a cell or a group of cells) or (ii) the inhabitants of an area.

In the first approach, the activity records are aggregated which take place in a given cell, regardless of which SIM cards produce them. In the case of the second version, those activity records are used that are produced by the inhabitants of the given cell, regardless of where the activity took place. The first approach can be called cell-based grouping and the latter, inhabitant-based grouping. The two approaches are illustrated in Figure 8. Cells can be grouped further to examine a larger area (residential, suburb, district, etc.).

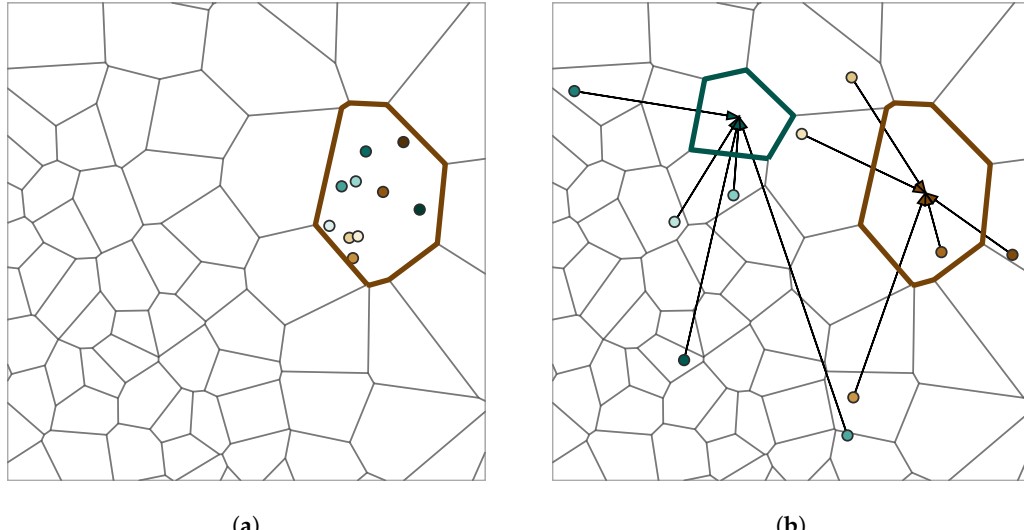

(**a**)　　　　　　　　　　　　　　　　　　　(**b**)

**Figure 8.** Visualizing the difference between the cell-based (**a**) and the inhabitant-based (**b**) approaches. The former considers subscribers present in a given cell wherever they live. The latter aggregates the activity of the inhabitants to the home cell, regardless of where the activity occurred.

## 4. Results and Discussion

In this section, the results are summarized and discussed, and the limitations and the future work are included.

### 4.1. Inhabitant-Based Approach

Based on the results of the "April 2017" data set (Figure 9a,b) the wake-up time almost always around 7:10 on workdays. On Sundays (or Easter Monday in the case of the long holiday), it is later by about one hour. On Saturdays (and the other days of the holiday), it is later by about 30–40 min, compared to an average workday.

The bedtimes are not so even, but also have a nice trend. The activity decreases between 19:40 and 20:10, on workdays, but during the holiday values, they are shifted by 30–50 min, similarly to the wake-up times. Interestingly, as bedtimes follows the wake-up times on the weekends, it results in the average day length remaining approximately the same. Figure 9c shows the day lengths, calculated as a difference of the bedtime and the wake-up time, in minutes. The day lengths are between 12 h 30 min and 13 h 10 min. On average, the day length is 12 h 45 min (the standard deviation is 10 min).

The same evaluation is performed on the "June 2016" data set. Figure 9d,e, shows the daily wake-up times and bedtimes, respectively. Basically, the same tendencies can be observed: wake-up times and bedtimes are shifted on holidays. However, there are some irregularities, especially within the bedtimes. On 22 June, the bedtime is almost on the same level as a weekend. The mobile phone activity decrease occurs 30 min later than on the other days of that week.

Figure 10 displays the inhabitant-based wake-up and bedtimes of the cells. This result is comparable to (Figure 9 in [39]); however, in that case, the wake-up times ($t_{wake}$) are earlier and bedtimes are later ($t_{sleep}$). Note that the difference ensues from the nature of the approach. In [39], the screen-on events of the phones are considered, while in Figure 10, the active usage of the mobile phone network is aggregated for a larger area. The "SensibleSleep" application [39] observes offline (from a mobile phone network perspective) smartphone activity that adjusts better to the actual SWC of an individual.

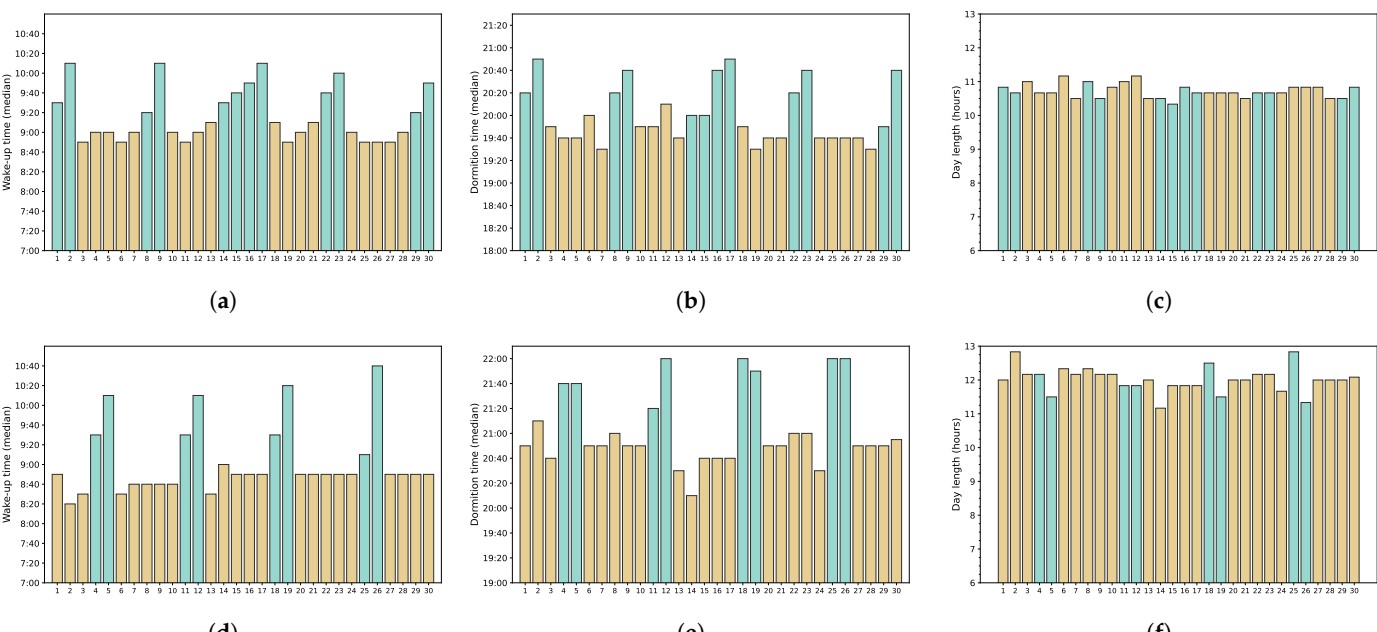

**Figure 9.** Daily wake-up times (**a**,**d**), bedtimes (**b**,**e**), and the day lengths (**c**,**f**), for the "April 2017" (**a**–**c**) and the "June 2016" (**d**–**f**) datasets. Holidays are represented by a different color.

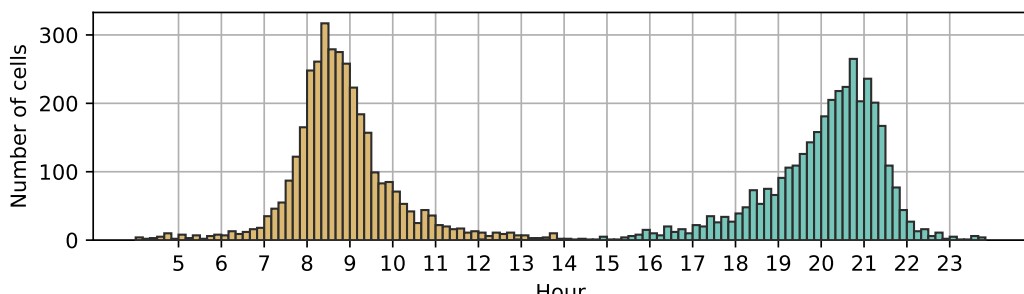

**Figure 10.** Inhabitant-based wake-up (brown) and bedtime (green) values of the cells, April 2017.

*4.2. The Length of the Day*

Figure 9a,b, shows that people start and end their days later on the holidays. This makes perfect sense, as they do not need to go to work, they do not need to spend time traveling to get somewhere in time, and they probably like to rest more. However, the bedtime is shifted as well, which results in the length of the day remaining practically the same; see Figure 9c.

Although the two data sets (Sections 2.1 and 2.2) were not recorded in the same year, they still cover two different months of a year (April and June). Considering that a mid-spring month is well represented by the "April 2017" dataset and an early summer month by the "June 2016" dataset (for Budapest), the differences between the wake-up time, bedtime, and the day length can be compared between the two seasons.

Figure 9d shows a small increase in the wake-up times, during the second half of the month. Although that 10 min increase in the averages cannot be considered significant, it may reflect the end of the school term (15 June). During the intersession, the time schedule of the public transport services is adjusted. For many lines, the headway is increased on workdays, while for some lines, it is decreased, especially on weekends. In this way, the intersession affects the whole transportation system of Budapest, and even its agglomeration to some extent.

Until the summer solstice (20 June 2016 [60]), the days are getting longer, but can the longer daylight have an effect on mobile phone network activity? As a reference, astronomical information (sunrise and sunset) is obtained from [61] for Budapest. Figure 11 shows the difference between the sunrises and the sunsets during the two observation periods of the data sets, projected to the same figure. The dashed lines display the values of June, and the solid lines display the values of April. As the summer solstice is in June, the differences during the month are negligible, but in April, the differences between the beginning and the end of the month are much more significant.

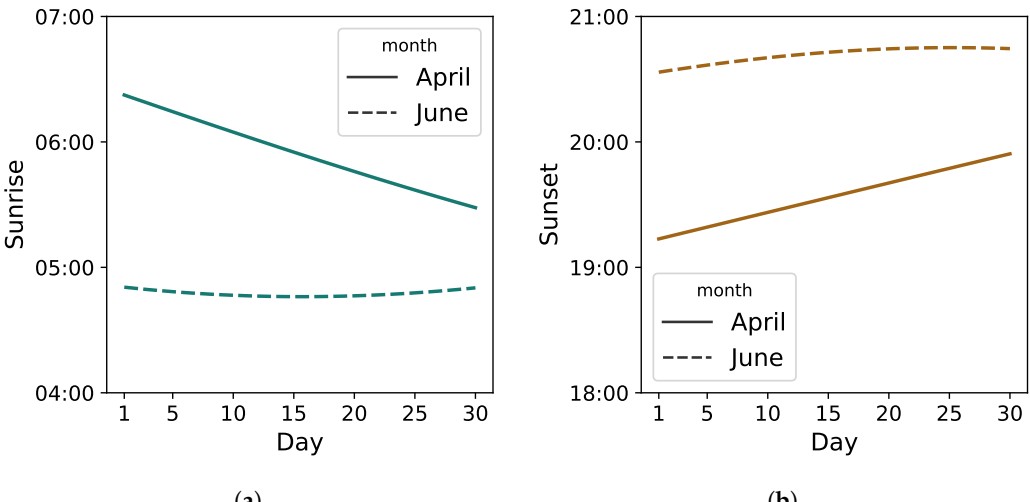

(a)                                                                                                        (b)

**Figure 11.** Sunrise (**a**) and sunset (**b**) times of June 2016 and April 2017 projected to the same figure to highlight the differences between the seasons, using data from [61].

So, in June, the sun rises earlier and sets later than in April, and consequently, lighter period of days are longer; 15 June is more than 2 h longer than 15 April, using the astronomical definitions of the sunrise and sunset. According to the calculated day lengths, the average workday length is 12 h in the "June 2016" data set, and 10 h and 40 min in the "April 2017" data set, resulting in a difference of 80 min.

Although, this is less than the astronomical difference, the sun rises very early in the morning, when people are still sleeping. It may be more practical to compare the results with the difference of the sunsets, as eople do not wake up earlier just to organize an activity before work, but may do so after work if it is still bright and the weather is good. The sun sets at 19:33 on 15 April and at 20:42 on 15 June, which is a 69 min difference, and is more comparable to the calculated values. The average workday bedtime values are 19:43 and 20:47, respectively. This finding—the day lengths are longer in June—is in good agreement with [47], where the seasonal influence of the daylight was examined via the length of the resting period.

The wake-up times are 8:57 and 8:44 on average (workdays), which show a slight decrease in the summer. Figure 9d shows slightly later wake-up times in the second half of the month. The average workday wake-up time in the first half of June 2016 is 8:39, and 8:50 in the second half. As mentioned in Section 4.1, it might be caused by the end of the school term.

### 4.3. Area-Based Approach

Based on the cell/area activity, the wake-up time and the bedtime were determined for every cell. Figure 12a,c shows the distribution for workdays, and Figure 12b,d shows the distribution for holidays, respectively. The results clearly show that the usage of the mobile phone network intensifies and reduces later on the holidays, indicating that people wake up and go to sleep later when they do not work.

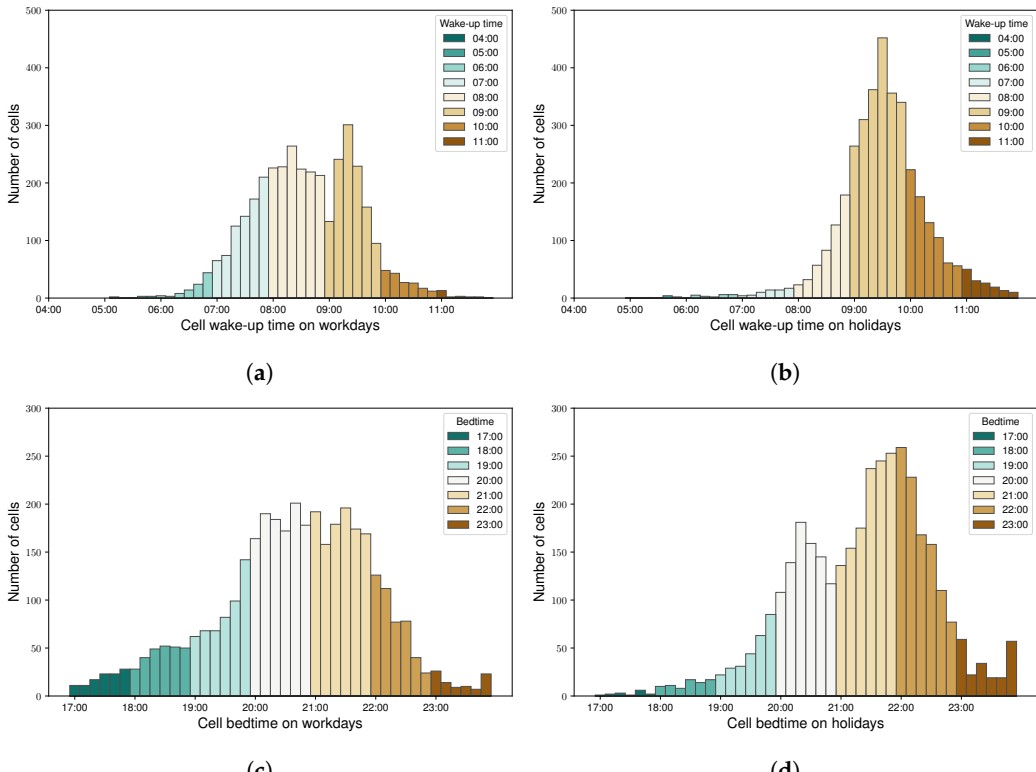

**Figure 12.** Cell based wake-up time (**a**,**b**) and bedtime (**c**,**d**) distribution for workdays and holidays, respectively, in the "April 2017" dataset.

Figure 13, shows the spatial distribution of the cell-based wake-up times. The Voronoi polygons, representing the mobile phone cells, are colored by the calculated "wake-up" times. The map clearly shows some extrema, where the times are around 10:00. Extrema 1–8 are all malls (WestEnd City Center (1), Arena Mall (2), Árkád (3), KÖKI Terminál (4), Lurdy Ház (5), Allee (6), MOM Park (7) and Mammut (8)) that uniformly open at 10:00, though 5 and 8 partly serve as office buildings.

Figure 14 shows the bedtime values of the sites, on workdays. As the Figure 12c illustrates with respect to cells, the bedtime is usually between 20:00 and 22:00. There are, however, some sites with a later bedtime, and a few of them are denoted in the figure. Marker 1 at the party district (Appendix A). In the site at marker 2, it does not have any distinctive object, which could explain this result. However, east of that site, there is a beer factory that might have notable activity in the evening—compared to the neighborhood— and the distortion of the Voronoi tessellation could have resulted in this late bedtime. At marker 3, there is a student's hostel, and, in the neighborhood, there are some sport and concert venues.

### 4.4. Working Hours

Figure 15 shows the activity distribution by days of the week and hours, based on the "April 2017" data set, separating the workplace (Figure 15a) and the home (Figure 15b) activity, but using the same scale. At the workplace, most of the activity is recorded during the working hours on workdays, as expected. The activity increases fast in the mornings, but decreases more slowly in the afternoon. The home activity is mostly clustered on the weekends, but after the work hours, in the evening, has a notable activity peak. The home activity before the working hours does not seem so significant.

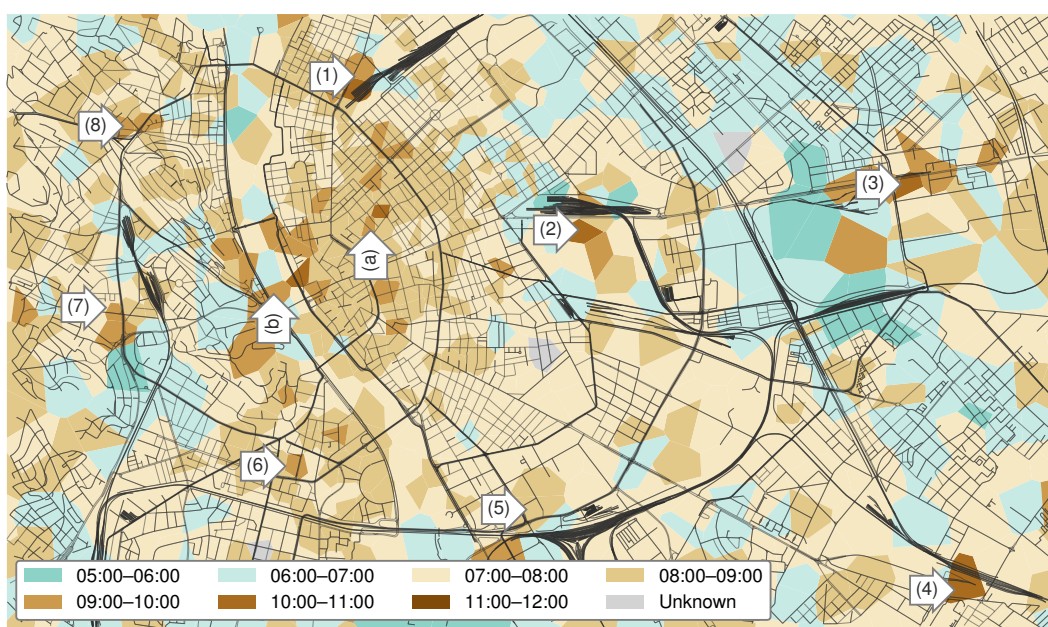

**Figure 13.** Spatial distribution of the cell-based wake-up times, malls (1–8) opens at 10:00.

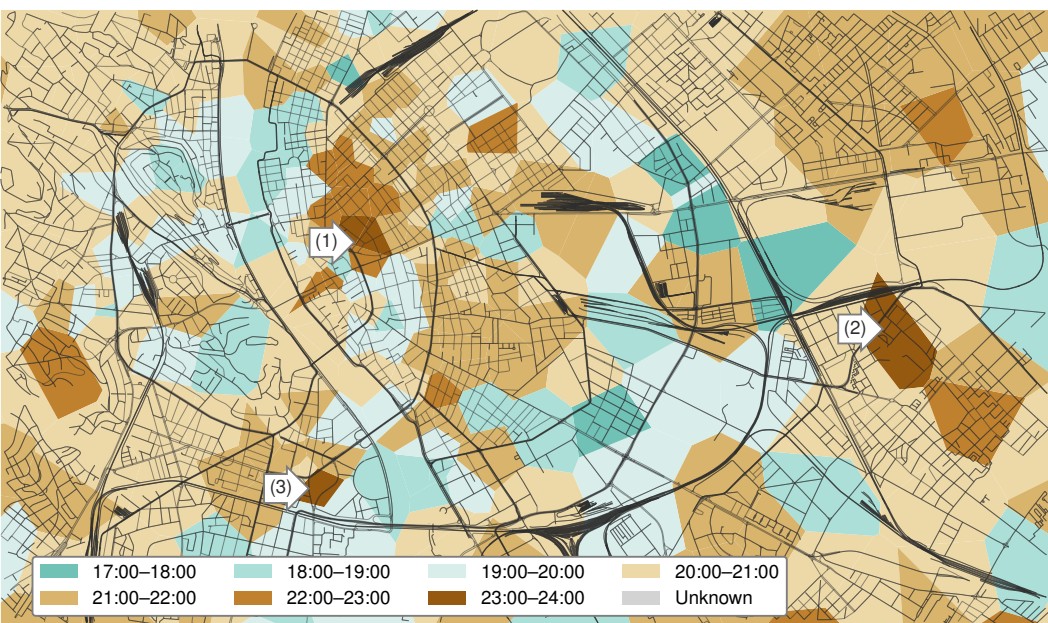

**Figure 14.** Spatial distribution of the cell-based bedtimes, aggregated to sites.

This procedure can be applied for every cell (or group of cells, such as sites), then the activity of the workers and the inhabitants will represent the workplace and home activity tendencies. Note that a cell can have both workers and inhabitants, so every cell has two aspects. Figure 16a,b illustrates the activity of the subscribers who work and live a selected site, respectively. Figure 16a also demonstrates the concept of Figure 7, using actual data.

As expected, Figure 16a,b is in accord with Figure 15a,b. The workers' activity increases in the morning and decreases late in the afternoon, whereas the inhabitants' activity decreases in the morning, increases late in the afternoon and reaches its peak in the evening. The two aspects of the same site have the opposite tendency.

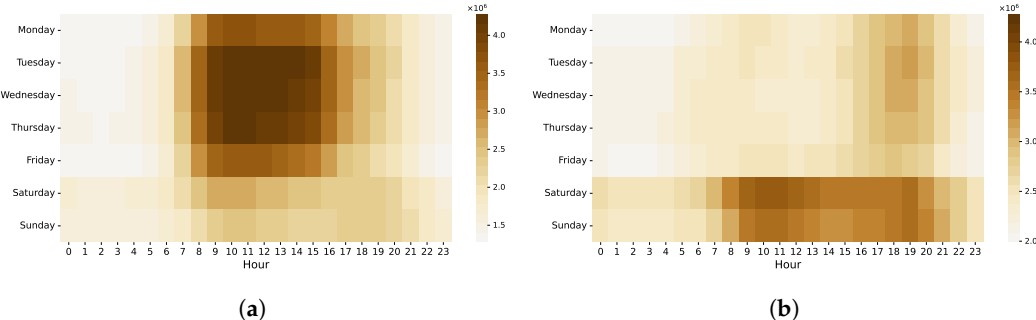

**Figure 15.** The mobile phone activity distribution by days of week and hours based on the "April 2017" data set, for the work places (**a**) and the homes (**b**).

Applying the same inhabitant-based approach as in the case of the wake-up time (Figure 7) to the workers' activity, it is possible to detect the positive and the negative edges of the activity curve. The positive edge could indicate the start of the working hours, and the negative edge could indicate the end of the working hours in a given cell (or site). Moreover, the difference of the two times can determine the length of the working hours.

In most of the sites, the working hours are about 8 h (Figure 17), just as expected. In the rest of the sites—especially where the working hour is less than 7 h or over 9 and a half hours—the mobile phone activity proves to be so low during the working hours that the results cannot be considered appropriate. Nevertheless, the mobile phone network reflects the length of the working hours.

The work hours do not necessarily start and end at the same time in every workplace. Are there any differences in this regard, from a mobile phone perspective?

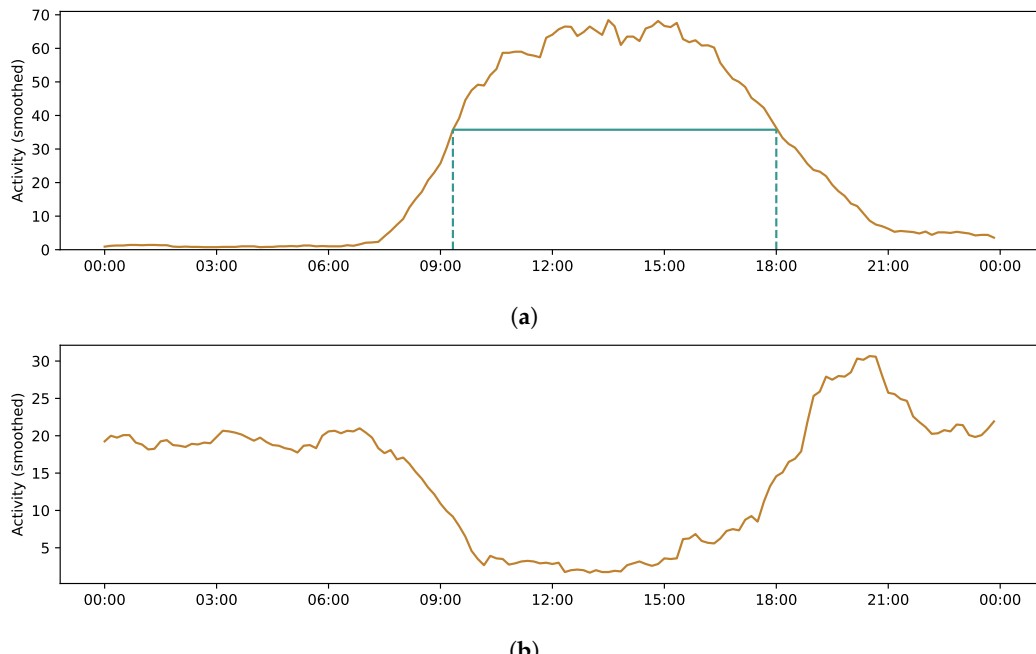

**Figure 16.** Activity of the workers (**a**) and the inhabitants (**b**) of a selected site, on a selected day.

Figure 17 shows the average workday length of the sites. The workday length in most of them are around eight hours, which agrees with the practice in Hungary. Moreover, the distribution of the working hour lengths also verifies the workplace detection approach, summarized in Section 3.1. Although there are some sites with very low (less than seven hours) or high (more than nine and a half hours) workday lengths, these sites have very few activities during the observation period: less than 2% of the total activity occurs in these sites.

The length of the working hours is usually around eight hours in most of the sites, but there might be differences at the beginning and the end of the labor time. Figure 18b shows the five most frequent beginning and ending labor times (defined as the positive and negative edges of the workplace activity curve of the site) and the connection between them, using a type of Sankey diagram. It is clear that in most of the sites, the mobile phone network activity increases between 8:30 and 9:30. That is not surprising, considering the lunch break. Note that the observed time values may have a delay compared to the actual start of the work, as an employee may not actively use the mobile phone network as soon as they arrive at their workplace (or when they leave it).

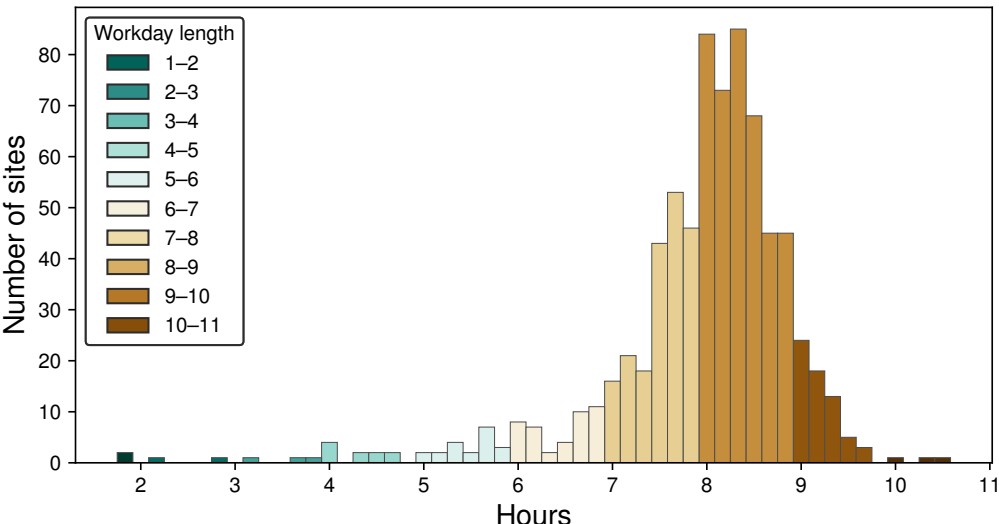

**Figure 17.** Distribution of workday length in sites.

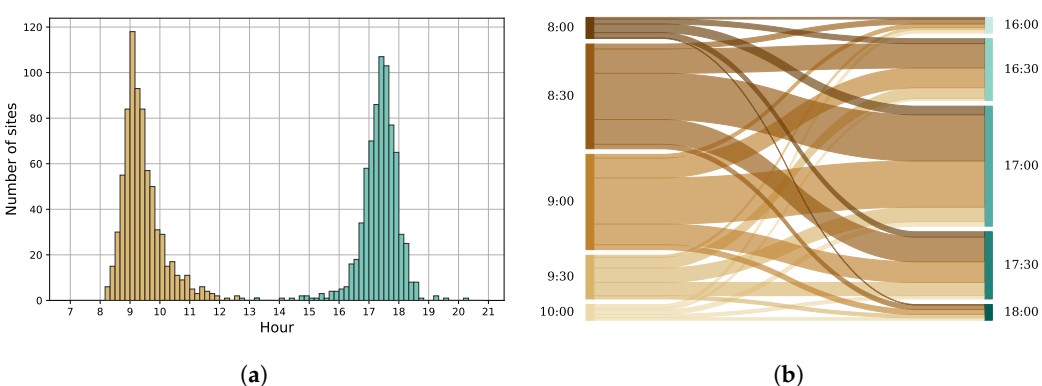

(a)           (b)

**Figure 18.** The distribution of the starting (brown) and the ending (green) of the working hours, in sites (**a**). The connection between the starting and the ending times (**b**).

### 4.5. With Respect to Mobility

In [27], we also showed that the mobility has a weekly seasonality. As Figure 9 shows, the wake-up time also has a seasonality. To compare it with the mobility indicators, described in Section 3.2, the daily wake-up times and the mobility metrics were normalized, using min-max feature scaling. Figure 19a,b displays the normalized entropy and the normalized radius of gyration—without the non-phone devices—in contrast to the wake-up times, respectively. As the mobility indicators are determined per subscriber, the inhabitant-based version of the wake-up time is used.

During workdays, both the entropy and the radius of gyration were high, but the wake-up time was low. On holidays, it is the opposite: the wake-up times were higher, and the mobility indicators were lower. This is not surprising, as people tend to wake up

earlier on workdays to go to work, as they usually need to travel to their workplace. On holidays, it is common to spend more time at home, which also reduces the mobility values. Figure 19 visualizes this clearly, especially in the case of entropy. Numerically (Pearson's R), the correlation between the wake-up times and the mobility indicators (entropy, radius of gyration) are $-0.8932$ and $-0.6873$, respectively.

Bedtimes show a similar trend (Figure 19c,d) in contrast to the two mobility metrics (entropy and radius of gyration), but the correlations numerically are not that strong: $-0.85$ and $-0.6621$, respectively. This might be caused by leisure activities after work. While people may go straight to work in the morning, they do not necessarily go home right after work.

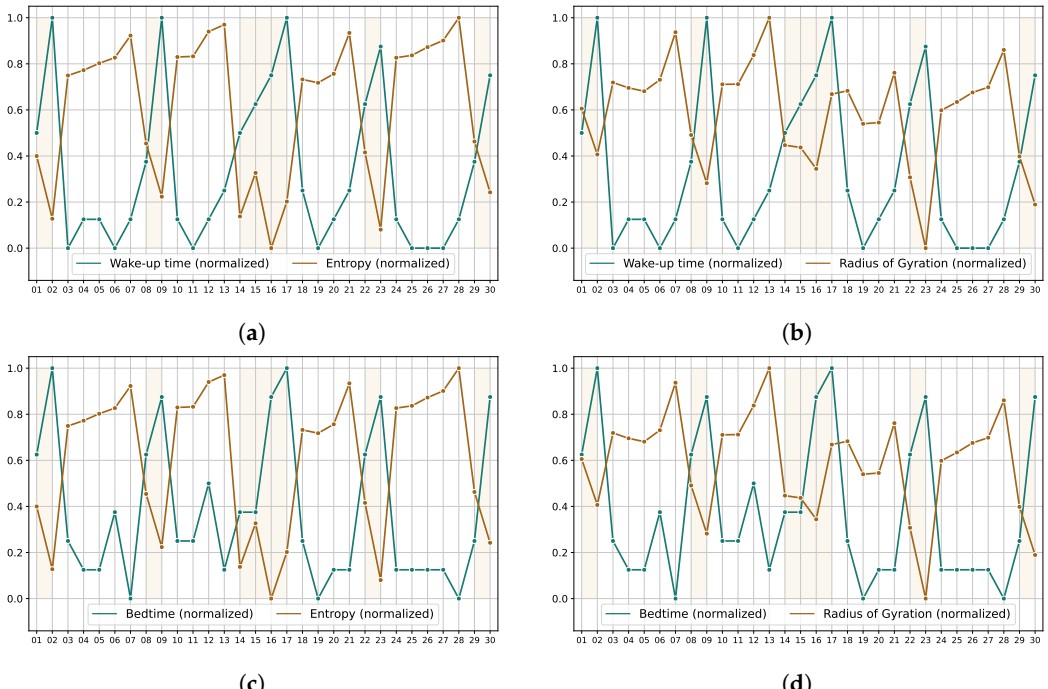

**Figure 19.** Normalized, inhabitant-based wake-up times in contrast of the normalized daily entropy (**a**) and radius of gyration (**b**). Pearson's Rs are $-0.8932$ and $-0.6873$, respectively. Second row: Normalized, inhabitant-based bedtimes in contrast of the normalized daily entropy (**c**) and radius of gyration (**d**). Pearson's Rs are $-0.85$ and $-0.6621$, respectively.

### 4.6. With Respect to Socioeconomic Status

In an earlier work [27], we demonstrated the correlations between the mobility customs, the distance of the home and work locations, and the socioeconomic status. We found that people who live in less expensive parts of Budapest tend to travel more to their workplace. The larger distance should indicate longer travel times and earlier wake-up times, as the work mostly starts at the same time in the morning. Lotero et al. previously found that "rich people do not rise early" [46], analyzing two cities of Colombia.

As for Budapest, there is a positive correlation between the socioeconomic status and the wake-up time. The inhabitant-based approach is used for this analysis, as the socioeconomic indicators are applied to subscribers. Besides the real estate prices [27], two properties of the mobile phones [4] were considered: the price and relative age. Figure 20a shows the wake-up times in contrast to the real property price and the mobile phone price categories. To give context to these categories, Figure 20b illustrates the number of subscribers in each category. Figure 21 has the same structure, but applying to the age of the cell phones.

Four property price categories and five phone price categories are formed. As Figure 6a shows, most of the price of one square meter in most of the real estate advertisements is under HUF 0.6 million. The property price categories are (i) HUF 0.3–0.5 million, (ii)

HUF 0.5–0.7 million, (iii) HUF 0.7–0.9 million and (iv) HUF 0.9–1.3 million. Mobile phone price categories are (i) EUR 0–150, (ii) EUR 150–300, (iii) EUR 300–450, (iv) EUR 450–600 and (v) EUR 600–750. As for the mobile phone age, five categories were formed from 0 to 5 years. The older phones were omitted. As Figure 6 indicates, most of the subscribers lived in less expensive homes and used less expensive cell phones, which were 1 to 3 years old.

In Figure 20a, the wake-up times are increasing in both dimensions. The lowest wake-up value is in the top-left corner, where the owners of the least expensive home locations and cell phones are grouped. Toward the bottom-right corner, the wake-up times are increasing. This observation clearly indicates that the richer people start their days somewhat later than the less wealthy people. The differences are numerically not large between the categories, but the observation area is also relatively small. Budapest is 525 square kilometers, and its diameter is about 30 km. The farthest part of the agglomeration is about 40 km from the city center.

Figure 21a reveals a negative correlation between the phone age and the wake-up time. This result implies that wealthier people do not only use more expensive phones, but tend to use the latest models. Presumably, they replace their phones more often.

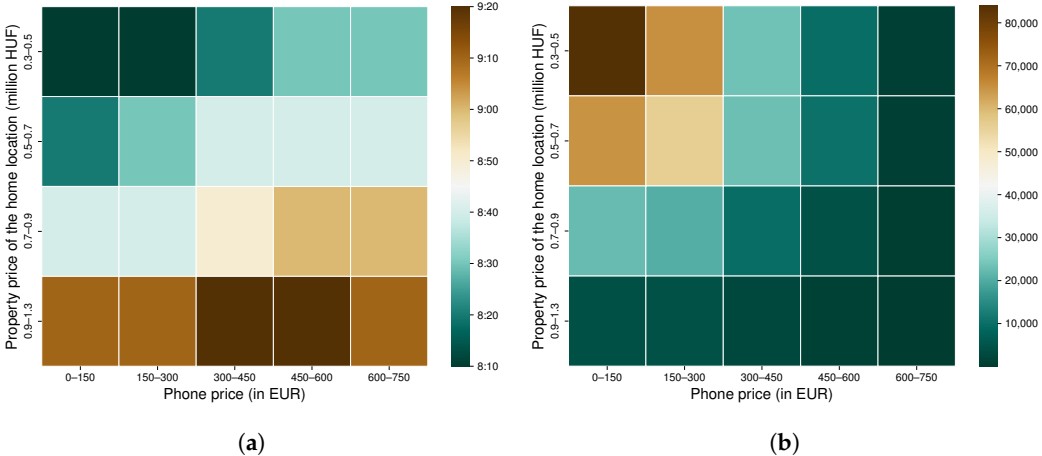

(a)                                                 (b)

**Figure 20.** The inhabitant-based wake-up times in the socioeconomic categories, based on the property price of the home location and mobile phone price (**a**), and the number of the subscribers in each category (**b**).

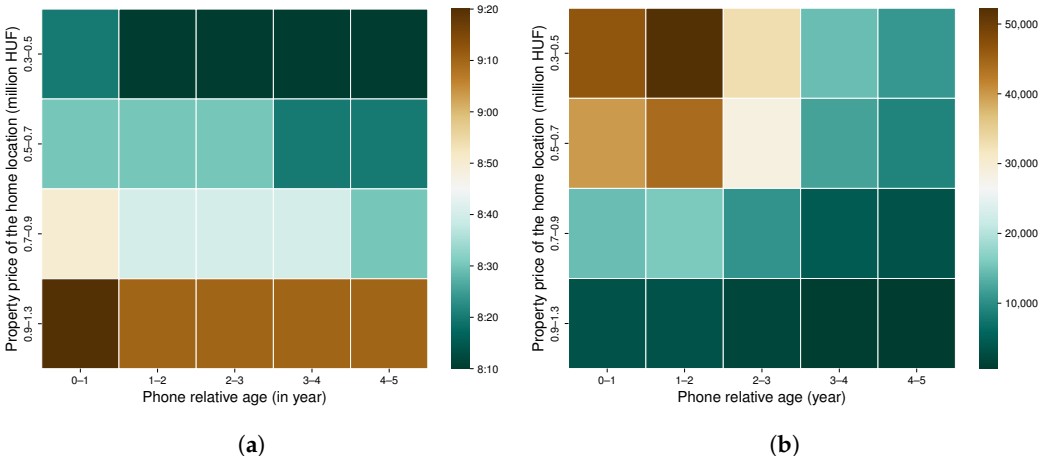

(a)                                                 (b)

**Figure 21.** The inhabitant-based wake-up times in the socioeconomic categories, based on the property price of the home location and the relative age of the mobile phone (**a**), and the number of the subscribers in each category (**b**).

*4.7. Limitations*

As the activity curves do not reach their maxima immediately, the selected time does not accurately represent neither the wake-up time, nor the time when the working hours actually start. These results can only be used relatively to other cells, or areas of the city. The reason for this is that the CDRs in this study only represent the active (in other word, billed) usage of the mobile phone network. However, the employees usually do not actively use the network at the moment they arrive to their workplace or home. There is a certain gap between that moment and the first activity. If the available data also contained the passive (cell-switching) communication, the time, when a SIM card enter the home or work cell, could be used. In this case, the terms "leaving home" and "arriving to the workplace" would be more accurate. Furthermore, the difference of the two could serve as a basis for a precise travel time estimation.

As the active mobile phone network activity is sporadic, the majority of the subscribers do not have enough activity records to trace back their daily movement accurately enough to determine when they leave their homes or arrive to their workplace. It is very common that the subscribers do not have activity at home before work. It would be necessary to have multiple activities in the morning to select the first and last home activities at home to conclude when the subscriber tends to wake up and leave their home. As this is not assured with this kind of data, this approach is not good for individual-level analysis.

The phone price database does not contain launch prices (see Appendix B), and the applied depreciation method is unknown. The results, utilizing the cell phone price as a socioeconomic indicator, should be interpreted by keeping that in mind.

*4.8. Future Work*

With the home and work locations, the distance can be determined; on the other hand, the travel time is hard to estimate. The subscriber could use different transportation modes that have different time demands. Furthermore, without cell-switching information, the time when the subscriber left the home cell and arrived to the work call cannot be exactly determined. Using the worker or inhabitant filtered activity curves—described in Section 4.4—the morning fall (and the evening rise) of the home location activity and the rise (and the fall) of the work location activity can be calculated. Considering that the work location activity rises when the workers usually arrive, and the home activity of the inhabitants drops when they usually leave for work, the difference of these values could be applied to estimate the travel time of a group of subscribers. This would, naturally, require a larger number of subscribers in every home–work cell pairs, or the locations should be aggregated by base stations or even larger areas of the city.

Although the "edge detection" method used in this paper gives reasonable results to determine the positive and negative edge of the activity curve, it would be worth comparing it with other approaches.

**5. Conclusions**

In this study, we introduced "wake-up time" as an indicator to describe the behavior of a group of subscribers. Due to the sporadic nature of the mobile phone network data, we grouped subscribers by home and work locations to analyze their activity at these locations. By determining the rising and the falling edges of the activity curve of the workers at each location, the beginning, the ending, and the length of the working hours were estimated. Tendencies between the starting and the ending time of the working hours at the work locations were also presented. This indicator is used not only to classify the groups of subscribers (inhabitant-based approach), but parts of Budapest (area-based approach), thus, city parts can also be characterized by chronotypes. This was demonstrated by real-life examples with the opening hours of the malls, in Budapest, or by the late activity fall of the party district.

The wake-up time as a proposed indicator was compared to common indicators, such as the radius of gyration or entropy, and a clear negative correlation was found. On workdays, both the radius of gyration and entropy values were higher, while the wake-up times were lower. On holidays, quite the contrary was found. The correlation between bedtime, the counterpart of the wake-up time, and the mobility metrics was not that strong, but still considerable.

The day length, calculated as a difference of bedtime and wake-up time, was found to be constant between the workdays and holidays: the start and the end of the day were also shifted. On the other hand, the day length reflected the seasonal differences of the two data sets: it was found that the days are longer, from the perspective of a mobile phone network, when there is more daylight. The longer days are the reason of the delayed activity fall, as the wake-up times were found to be marginally affected by the earlier sunrise.

Using socioeconomic classes derived from housing prices at the home location, mobile phone prices and the age of the cell phone, the correlation between the wake-up time and the socioeconomic status was also identified. The subscribers living in less expensive apartments get up earlier, and this tendency holds true with respect to mobile phones prices: subscribers who own more expensive cell phones tend to get up later.

These results may help to analyze further the city structures, by identifying "early bird" or "night owl" areas and the possible connections between them. City parts with early morning or late night activities may require different public transport services, for example, and can aid transportation infrastructure planning. Business development could also benefit from the detailed insight of the neighborhood chronotypes, especially with the associated information of the home locations and the socioeconomic status of the subscribers.

The socioeconomic aspect of these findings can also contribute to better understanding the social structure of an urban environments. In this regard, further studies need to be made, possibly with detailed census data, to evaluate the commuting and working habits of the different socioeconomic classes.

**Author Contributions:** Conceptualization, G.P.; methodology, G.P.; software, G.P.; validation, G.P.; formal analysis, G.P.; investigation, G.P.; resources, G.P. and I.F.; data curation, G.P.; writing—original draft preparation, G.P.; writing—review and editing, G.P.; visualization, G.P.; supervision, I.F.; project administration, I.F.; funding acquisition, I.F. All authors have read and agreed to the published version of the manuscript.

**Funding:** This research supported by the project 2019-1.3.1-KK-2019-00007 by the Eötvös Loránd Research Network Secretariat under grant agreement no. ELKH KÖ-40/2020. The authors acknowledge the financial support of this work by Hungarian-Japanese bilateral project (2019-2.1.11-TÉT-2020-00204).

**Institutional Review Board Statement:** Not applicable.

**Informed Consent Statement:** Not applicable.

**Data Availability Statement:** CDR and TAC data, used this study, are not publicly available due to third party restrictions.

**Acknowledgments:** The authors would like to thank Vodafone Hungary and 51Degrees for providing the Call Detail Records and the Type Allocation Code database used in this study. The estate price data are provided by the ingatlan.com estate selling portal. For plotting the maps, OpenStreetMap data were used; these data are copyrighted by the OpenStreetMap contributors and licensed under the Open Data Commons Open Database License (ODbL).

**Conflicts of Interest:** The authors declare no conflict of interest. The funders had no role in the design of the study; in the collection, analyses, or interpretation of data; in the writing of the manuscript; or in the decision to publish the results.

## Abbreviations

The following abbreviations are used in this manuscript:

| | |
|---|---|
| CDR | Call Detail Record |
| HUF | Hungarian forint |
| IMEI | International Mobile Equipment Identity |
| OSM | OpenStreetMap |
| POI | Point of interest |
| SIM | Subscriber Identity Module |
| SES | Social Economic Status |
| SWC | Sleep Wake Cycle |
| TAC | Type Allocation Code |

## Appendix A. Party District

The so-called party district is not an official area of the city with definite borders. Usually, the area, bounded by the Károly Boulevard (iii), Király Street (v), Erzsébet Boulevard (i) and Rákóczi Street (ii), is referred to as the "party district" (see Figure A1a). This area is famous for ruin pubs and nightlife, which also involves the fact that it is not the most active area early in the morning. In fact, the vivid nightlife of downtown is not confined only to this area. The Deák Ferenc Square (iv), the northern side of the Király Street (v), and the eastern side of the Erzsébet Boulevard (i) can also be considered part of the party district.

How big is the party district? To answer this question, the points of interest (POI) were downloaded from OpenStreetMap (OSM) in the categories of "bar", "pub", "biergarten" and "nightclub", using the OSM terminology, and plotted to a map (see Figure A1b). There are 192 bars, 472 pubs, 12 beer gardens and 31 nightclubs in the displayed, 9 km by 9 km part of Budapest, and the area from Figure A1a is also highlighted. Note that the POIs are queried in 2021, and cannot show the exact state of 2017. However, only the tendencies are important, which have not changed fundamentally in the last decade. The concentration of bars and pubs is, indeed, the highest within the highlighted "party district" area, but still very high on the whole Pest-side of the downtown, and there are also three smaller groups on the Buda-side of the city.

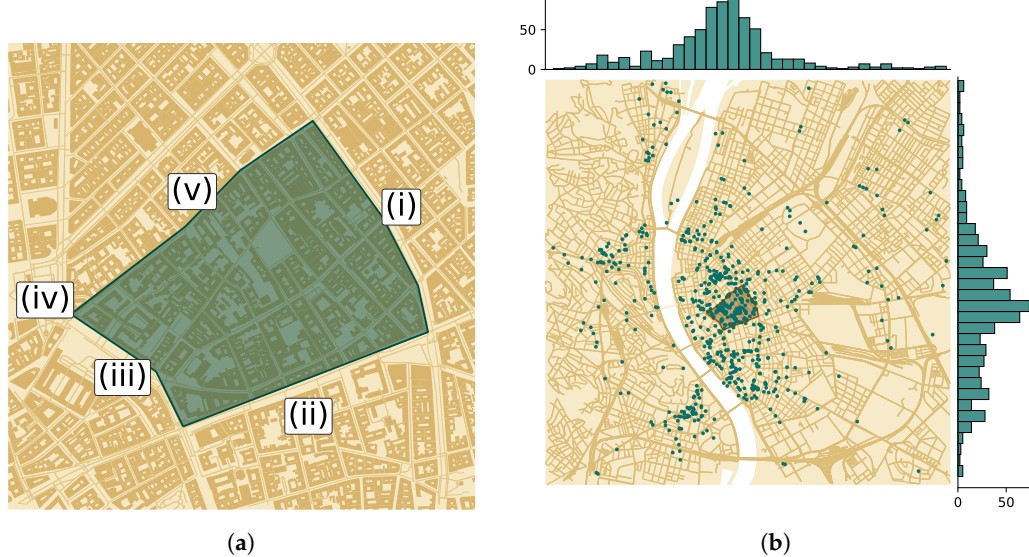

(**a**)                                                                                    (**b**)

**Figure A1.** The borders of the party district (**a**), within District 7; and bars, pubs, beer gardens and nightclubs in Budapest downtown (**b**), based on OpenStreetMap data.

## Appendix B. iPhones

As Apple iPhones are considered a status symbol [62], it makes them suitable to validate the phone price database [56], described in Section 2.3. Figure A2a shows the

number of subscribers that exclusively use certain iPhone models in the "April 2017" dataset. Using TAC values, it is not possible to distinguish the iPhone models based on a specification such as storage. However, it is clear that the most expensive models ("Plus" versions) do not have a significant user base, in contrast with some older models, such as the iPhone 4 and iPhone 5 series.

The launch prices of the iPhone models, released until April 2017, were obtained from [63]. Figure A2b compares the two sources. As there are different versions of the certain models, "budget" (with the lowest amount of storage) and "high-end" (the most expensive) versions are also displayed. Although the GSMArena price property is supposed to be a launch price, Figure A2b clearly shows that they are much lower than the original prices. Moreover, the older the phone is, that lower the available prices are, except for the first iPhone. It has to be noted that GSMArena prices are in EUR, whereas the ground truth prices are in USD, but this cannot cause the difference. The results of this analysis implies that the phone prices might have depreciated.

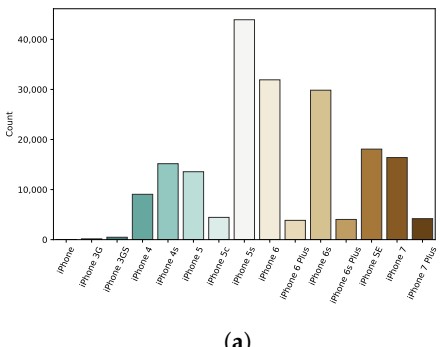
(a)

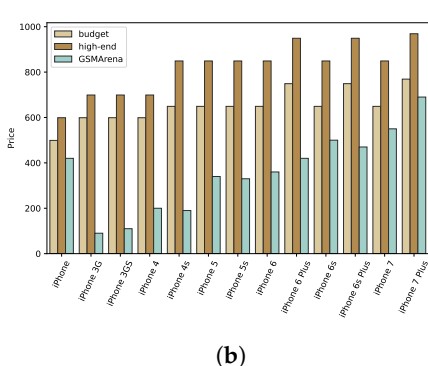
(b)

**Figure A2.** Based on the "April 2017" dataset, the different iPhone models in use are also displayed (**a**), and comparing Apple iPhone prices [63] with the GSMArena-based source [56] (**b**). Versions with the lowest amount of storage are denoted by "budget", and versions with the most expensive versions are categorized as "high-end".

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
