# Peer review of "Awakening City: Traces of the Circadian Rhythm within the Mobile Phone Network Data"

_information, doi:10.3390/info13030114_

Round 1

Reviewer 1 Report

This paper presents a method of using the mobile data to determine the wake-sleep cycle and the level of activity of people in the area of Budapest. The presentation is of very high quality and the work is generally sufficiently explained. Some issues with the current work that ought to be addressed are: —figure 7 refers to the calculation of the activity time of the subjects but no equation describing this calculation has been presented. —the conversion of TACs to prices for the phones (section 2.3) using GSMarena and 51degrees seems to hold sense initially, but certainly more information about the efficacy and validity of this method need to be provided. Given the stats of cellphones in Hungary in 2017, one sees the percentages of devices running iOS being significantly higher than what the figure 6b suggests they ought to be. Their average price is at the top or higher of the entire 6b spectrum, and they possess a 10% market share with the most conservative estimates (17% according to some readily available online resources), something which is not represented in this graph. More explanations are needed in these correlations for the socioeconomic indicators. —in the subsections 4.4 and below it should be made clear whether the inhabitant based or area based approach were being used for each of the presented results. Overall I suggest that this paper is published with minor revisions addressing the above.

Author Response

1. The equation from Figure 7 is now presented in Equation 3. See also, lines 231-233.

2. Mobile phone prices has been analysed further. Appendix B, contains an analyses about the iPhones, that were in use during April 2017.
Figure A2a shows the number of SIMs working in different iPhone models, Figure A2b compares the GSMArena prices to the real iPhone prices.
It seems that the GSMArena source contains depreciated prices, as the older the phone, the larger the difference in price from the ground truth.

The prices were not changed, but I believe that this validation and more detailed description make the application clearer.

A warning is also added to the limitations, see line 483-485.

Section 2.3, more specifically lines 183-194, has been also modified, as the relative age of the phones are also discussed. Figure 6c displays a histogram.

Age of the phones are also compared to the wake-up times and a negative correlation id found. See Figure 21 and lines 459-461, but Section 4.6 contains other modifications regarding this new figure.

3. subsection 4.4 and below uses the inhabitant based approach, that has been made clear
line 378, lines 413-414, Figure 19 caption, lines 436-437, Figure 20 and 21 captions.

Reviewer 2 Report

It is a good, well-written paper, based on an extensive and complex quantitative analysis, and providing interesting results. The literature review is comprehensive, the method is clearly described, and the results are discussed extensively. The section that needs further development is the Conclusion. There is no clearly identified value added of this article, or stakeholders that may benefit from these results.

On a minor note, although the English language and style of the manuscript are very good, the authors should check again the text to make some statements clearer. See for instance the alternative verbs in Line 415 “Both the real estate prices and the mobile phone prices [4] were used considered” and Line 438: ‘However, the employees usually do not use actively use the network …”.

Author Response

The explicitly mentioned language error has been fixed, see line 439 (the whole sentence is modified because of other requests in the section) and line 469.

And some other changes has been made to improve the text quality.

Beside some minor changes, a new paragraph is added to the conclusion to address the mentions weakness. See lines 527-537.

Reviewer 3 Report

What I miss in the paper is the analysis of works from Barabasi, which discusses and analyzes human mobility patterns also based on CDR data, and discusses the Levy flight characteristics of human mobility and the difference between this approach and the real mobility data.

  1. Gonzalez M. C., Hidalgo C. A., Barabasi A. L.: Understanding individual
    human mobility patterns, Nature, vol. 453, s. 779—782, Jun.
    2008.
  2. Song C., Qu Z., Blumm N., Barabasi A.: Limits of Predictability in
    Human Mobility, Science 327, 1018, 2010.
  3. Barabasi A. L.: The Origin of Bursts and Heavy Tails in Human Dynamics,
    Nature, s. 207–211, 2005.

Also would it be possible to release the anonymised data set for other scientists to use? I believe that would be quite helpful in e.g. designing more accurate human mobility models.

Author Response

The mentioned papers are now cited, see lines 66-69, or reference 35-38.
However, I believe Lévy-flight is not relevant for this study in more details. It is because, Lévy-flight is an individual-level characteristic. (We previously found evidence of this property in our data.) But, in this study we have analysed subscribers on an aggregated level. Section 3.4, details two different aggregation methods, that were applied. Though, this is partly because of the nature of our concrete data set.

As for the data sets, both Vodafone Hungary and 51Degrees provided exclusive access to these data for Óbuda University, so we cannot share it.
If I may add, I'd be happy to publish it on Zenodo or other similar platform, but I cannot do that.

Reviewer 4 Report

In this work, the authors use CDR to analyze the circadian rhythm of the subscribers. The topic is interesting however the paper should be improved.

The organization and structure of the paper should be improved. Introduction should be rewritten. It does not provide an overview of current knowledge, allowing researchers to identify relevant existing research gaps.

It would be better to demarcate the Introduction and the survey of the related work as much as possible. The authors are advised to give more attention to the research objective, the methodological approach, and the research challenges in the Introduction. Eventually, it would help the reader to see and comprehend the intent, contribution, and novelty of the work.

Many references that should have been cited are missing. The following papers are relevant:

M. Ghahramani, M. C. Zhou, Y. Qiao and N. Q. Wu, "Spatio-Temporal Analysis of Mobile Phone Network based on Self-Organizing Feature Map," in IEEE Internet of Things Journal.

M. Ghahramani, M. Zhou and C. T. Hon, "Extracting Significant Mobile Phone Interaction Patterns Based on Community Structures," in IEEE Transactions on Intelligent Transportation Systems.

Peng, L. Liu and L. Zhang, "A Hive-Based Retrieval Optimization Scheme for Long-Term Storage of Massive Call Detail Records," in IEEE Access,

In the Conclusion part, what the concrete scientific propositions are? I

Future research perspectives should be enhanced from different angles. I think the paper has some publishable material, and I encourage the authors to revise and resubmit the paper.

Author Response

With all due respect, but I think that a complete rewrite of the Introduction is not necessary. After some reconsideration, we have added another paragraph to the very beginning of the article (see lines 18-24) to introduce the basis of this work better.

Then, the literature review is used to introduce the field in more depth. The topics of the survey are chosen to focus the field to the main topic of this paper. Namely, (i) social sensing (with a very recent COVID-19 applications): lines 32-45, (ii) commuting analysis: lines 46-51, (iii) socioeconomic aspect of the CDR data processing: lines 52-63 and (iv) sleep-wake-cycle/circadian rhythm analysis (lines 70-85), which is the main topic.

This is also important, because -- based on this -- the mentioned papers are simply not relevant enough to discuss.

Peng et al.: "This paper proposed a novel DCB scheme, in which a Hive-based extended hash storage, other than the traditional sequential storage, is employed." Where DCB means dual-column bucketing, so this paper is completely irrelevant. As in our paper we do not present anything even broadly related to that. Though it is a fine paper, I daresay, that the only connection is the CDR data.

Ghahramani (2021): In this paper, Self-Organizing Feature Map is applied to cluster Macau, utilizing spatio-temporal properties. Figure 9 (of that work) summarized this very nicely. However, it can also be seen, that the proposed approach very far from ours, as the details of the timeseries are not considered in this approach. It is clear, that different regions of Macau has different magnitude of mobile phone traffic, but no finer detail is presented about the daily activity. As opposed to our approach, where the start and the end of the daily activity peaks (in respect of very small areas) are studied to infer mobility and socioeconomic features of a group of subscribers.

Ghahramani (2018): I have cited this paper before (https://www.mdpi.com/2078-2489/12/11/468), where commuting was more broadly discussed. In this paper, the very first step is to determine the home and work locations (Section 3.1). None of the mentioned papers do this. 
There are many other paper that could have been cited, but I intended to cite only the most relevant ones. This paper has, on the other hand, an interesting finding about people tend communicate within their communities, so I found an opportunity to cite it (reference 28), without restructuring the survey part of the introduction.
Though, no socioeconomic feature is included. It would be interesting if they extend their research in that direction, regarding Macau.

After these, in a form of a coupe of questions, the motivation and goals are detailed (line 86-90), finally the contributions are presented: lines 93-98.

As for the Conclusions, a new paragraph is added to the conclusion to address the mentioned weakness. See lines 527-537. And there were some minor changes as well, in that section.

Round 2

Reviewer 4 Report

I am happy with this version and I believe the paper is now ready for publication